# Discovery of a Gatekeeper Residue in the C-Terminal Tail of the Extracellular Signal-Regulated Protein Kinase 5 (ERK5)

**DOI:** 10.3390/ijms21030929

**Published:** 2020-01-31

**Authors:** Adam J. Pearson, Paul Fullwood, Gabriela Toro Tapia, Ian Prise, Michael P. Smith, Qiuping Xu, Allan Jordan, Emanuele Giurisato, Alan J. Whitmarsh, Chiara Francavilla, Cathy Tournier

**Affiliations:** 1Division of Cancer Sciences, School of Medical Sciences, Faculty of Biology, Medicine and Health, The University of Manchester, Manchester M13 9PT, UK; adam-pearson1@hotmail.co.uk (A.J.P.); qiuping.xu@postgrad.manchester.ac.uk (Q.X.); emanuele.giurisato@manchester.ac.uk (E.G.); 2Division of Molecular and Cellular Function, School of Biological Sciences, Faculty of Biology, Medicine and Health, The University of Manchester, Manchester M13 9PT, UK; Paul.Fullwood@manchester.ac.uk (P.F.); gabytatolina@gmail.com (G.T.T.); michael.smith-8@manchester.ac.uk (M.P.S.); Alan.J.Whitmarsh@manchester.ac.uk (A.J.W.); chiara.francavilla@manchester.ac.uk (C.F.); 3Division of Infection, Immunity and Respiratory Medicine, School of Biological Sciences, Faculty of Biology, Medicine and Health, The University of Manchester, Manchester M13 9PT, UK; ian.prise@postgrad.manchester.ac.uk; 4Drug Discovery Unit, Cancer Research UK Manchester Institute, The University of Manchester, Manchester M13 9PT, UK; a.jordan@sygnaturediscovery.com; 5Department of Biotechnology Chemistry and Pharmacy, University of Siena, 53100 Siena, Italy

**Keywords:** ERK5, MAPK, phosphorylation, proteomics, mass spectrometry, transcription

## Abstract

The extracellular signal-regulated protein kinase 5 (ERK5) is a non-redundant mitogen-activated protein kinase (MAPK) that exhibits a unique C-terminal extension which comprises distinct structural and functional properties. Here, we sought to elucidate the significance of phosphoacceptor sites in the C-terminal transactivation domain of ERK5. We have found that Thr^732^ acted as a functional gatekeeper residue controlling C-terminal-mediated nuclear translocation and transcriptional enhancement. Consistently, using a non-bias quantitative mass spectrometry approach, we demonstrated that phosphorylation at Thr^732^ conferred selectivity for binding interactions of ERK5 with proteins related to chromatin and RNA biology, whereas a number of metabolic regulators were associated with full-length wild type ERK5. Additionally, our proteomic analysis revealed that phosphorylation of the Ser^730^-Glu-Thr^732^-Pro motif could occur independently of dual phosphorylation at Thr^218^-Glu-Tyr^220^ in the activation loop. Collectively, our results firmly establish the significance of C-terminal phosphorylation in regulating ERK5 function. The post-translational modification of ERK5 on its C-terminal tail might be of particular relevance in cancer cells where ERK5 has be found to be hyperphosphoryated.

## 1. Introduction

Mitogen-activated protein kinase (MAPK) pathways are evolutionarily conserved signaling cascades involved in the regulation of a variety of fundamental cellular processes such as cell proliferation, differentiation, survival and migration [1]. These pathways comprise a three-tier protein kinase cascade which results from the sequential activation of a MAPK kinase kinase (MAPKKK), a MAPK kinase (MAPK) and a MAPK, to relay, amplify and integrate many different stimuli, including mitogens, cytokines and various environmental stresses. MAPKs mediate responses to these extracellular cues mainly by regulation of gene expression, but also by post-translational mechanisms involving cytoplasmic targets [1]. The best-characterized members of the MAPK family are the extracellular signal-regulated protein kinases (ERK) 1 and 2, the c-Jun N-terminal protein kinases (JNK) and the p38 MAPKs. In contrast, ERK5 is much less understood despite compelling genetic evidence that ERK5 exerts non-redundant and important function during development and in various diseases [2]. ERK5 was first identified in 1995 by two independent research groups who immediately recognized the comparatively large size of this novel MAPK [3,4]. Specifically, ERK5 exhibits an apparent molecular weight of approximately 100 kDa due to a unique C-terminal extension not present in other members of the MAPK family. The C-terminal tail of ERK5 contains two proline-rich domains (PR1 (aa 434-464) and PR2 (aa 578-701)), a myocyte enhancer-binding factor 2 (MEF2)-interacting domain (aa 440-501), a nuclear localization signal (NLS (aa 505-539)) and a transactivation domain (TAD (aa 664-789)) [5]. 

In quiescent cells, ERK5 exists in a “closed” folded conformational state defined by an intermolecular interaction between the N- and C-terminal half which constitutes a nuclear export signal (NES) [6,7]. In this conformation, the majority of ERK5 is sequestered in the cytoplasm. In response to stimulation, the MAPK/ERK kinase 5 (MEK5) phosphorylates the canonical MAPK activation motif Thr^218^-Glu-Tyr^220^ in the N-terminal domain of ERK5. This disrupts the intermolecular interaction and subsequently induces a conformational change to an “open” state, and exposure of the NLS [6,7]. In addition to controlling nucleo-cytoplasmic shuttling, the phosphorylation of ERK5 by MEK5 increases ERK5 catalytic activity, thereby enhancing transcription via the phosphorylation and activation of downstream transcription factors, including members of the MEF2 family [5]. More surprisingly, ERK5-mediated transcription is also controlled by autophosphorylation at various phosphoacceptor sites identified in the C-terminal TAD [8]. This domain was originally identified by evidence that C-terminal truncated mutations impaired ERK5 transcriptional activity [9,10]. Recent discoveries that the C-terminus of ERK5 was phosphorylated by cyclin-dependent protein kinase (CDK) pathways during mitosis [11,12] and by ERK1/2 in response to growth factor stimulation [13] revealed unique potential cross talk mechanisms underlying ERK5-mediated transcription, independently of MEK5. 

Together, these studies provided strong evidence that phosphorylation of the C-terminal tail of ERK5 was functionally important. This finding is of particular significance in human epidermal growth factor receptor 2 (HER2) over-expressing breast cancer cells which exhibit a hyperphosphorylated form of ERK5 [14]. Remarkably, over-expression of ERK5 in this breast cancer subtype correlates significantly with poor prognosis [15]. Additionally, C-terminal phosphorylation of ERK5 was shown to be enhanced in melanoma cells expressing oncogenic BRAF^V600E^ [16]. In light of this knowledge, this study details our elucidation of the mechanism underlying ERK5-dependent transcription via post-translational modification of its C-terminal tail. Given the paucity of information on molecular interactions underpinning ERK5 signaling, our approach was based on high throughput comparative analyses of C-terminal phospho-ERK5 mutants created by site-directed mutagenesis. Our results identified a gatekeeper residue in the C-terminal portion that modulates the signaling function of ERK5 by influencing the binding selectivity of ERK5 with downstream targets. 

## 2. Results

### 2.1. Thr^732^ is Critical for C-Terminal Phosphorylation of ERK5

Detailed phosphomapping analysis of ERK5 provided the first evidence that ERK5 could autophosphorylate its tail [17]. Further studies confirmed that the TAD in the C-terminal region of ERK5 contained several putative phosphoacceptor sites, including: Ser^706^, Ser^719^, Ser^730^, Thr^732^, Ser^753^, Ser^769^, Ser^773^ and Ser^775^ [8,11,12,13]. Importantly, the data showed that C-terminal phosphorylation controlled the ability of ERK5 to translocate to the nucleus and to regulate transcription [11,12,13]. To address the fundamental basis of this unique feature, we generated various ERK5 mutants using a site-directed mutagenesis approach. cDNAs encoding Flag epitope-tagged human ERK5 full length (ERK5-FL), truncated and phospho-mutants, were subsequently integrated into the genome of HeLa cells using the Flp-In T-Rex system to create isogenic inducible stable recombinant cell lines (Figure 1A). ERK5 expression was induced by incubating the cells with tetracycline. 

Initially, we confirmed that ERK5-FL induced in cells expressing a dominant active mutant form of MEK5 (MEK5D) or in cells stimulated with epidermal growth factor (EGF), exhibited a mobility shift by SDS-polyacrylamide gel electrophoresis (SDS-PAGE) characteristic of post-translational modification by phosphorylation (Figure 1B). ERK5 mutants lacking the C-terminal tail (ERK5-ΔC) did not display this striking electrophoretic behavior (Figure 1B,C). In contrast, the analysis of a C-terminal fragment, namely ERK5-ΔN, comprising the PR1, PR2 and NLS, revealed distinct mobility shifts under basal or stimulated conditions, indicative of multiple constitutively phosphorylated residues in the C terminus of ERK5 (Figure 1C). We further analyzed the migration pattern of two phosphomutants, ERK5-4xA_i_ and ERK5-4xA_ii_, in which four conserved putative phosphoacceptor sites in the TAD (i.e., Ser^706^, Thr^732^, Ser^753^ and Ser^773^, or Thr^732^, Ser^769^, Ser^773^ and Ser^775^) were replaced by non-phosphorylatable alanines (Figure 1A). ERK5-4xA mutants migrated with a slightly accelerated mobility compared with ERK5-FL due to the lower molecular weight of Ala compared with that of Ser/Thr residues. The demonstration that these substitutions caused the disappearance of the slow migrating band in EGF-treated cell extracts confirmed that C-terminal phosphorylation was responsible for the electrophoretic migratory shift (Figure 1D). Likewise, the shifted ERK5-FL band was not detected in EGF-treated cells pre-incubated with XMD8-92, supporting the idea that activated ERK5 autophosphorylates its tail (Figure 1E) [8,17]. Thr^732^ is the only proline-directed phosphoacceptor site in the TAD of ERK5 that has been consistently found phosphorylated [8,11,12,13,17]. Remarkably, single substitution of Thr^732^ by alanine (T732A) caused the disappearance of the electrophoretic shift upon EGF stimulation (Figure 1D,E). Kinetic analysis of EGF treatment confirmed that T732A mutation impaired, rather than delayed, ERK5 C-terminal tail phosphorylation (Figure 1F). Collectively, these data indicated that phosphorylation at Thr^732^ was a critical step in the post-translational modification of ERK5. 

### 2.2. Phosphorylation at Thr^732^ Promotes ERK5 Nuclear Translocation

Previous studies have demonstrated that C-terminal phosphorylation of ERK5 was important for controlling the subcellular localization of ERK5 [11,12,13]. Initially, we confirmed that, under basal conditions, ERK5-FL predominantly resided in the cytoplasm, whereas ERK5-ΔC (1-575) was mainly detected in the nucleus (Figure 2A,B). This was consistent with the nuclear export-dependent activity previously defined by an intermolecular interaction between the N- and C-terminal half of the protein, and the presence of a functional NLS (aa 505-539) in the C-terminal region [6,7,18]. Accordingly, the constitutively phosphorylated ERK5-ΔN (411-816) fragment displayed an exclusive nuclear localization (Figure 3). Interestingly, the ERK5-ΔC (1-408) and ΔC (1-503) mutants which lack both the nuclear export-dependent activity and the NLS (Figure 1A), remained strongly nuclear (Figure 3). To explain this observation, we propose that the C-terminal tail of ERK5 might comprise a cytoplasmic retention function, possibly through protein association, that contributes to controlling the dynamic shuttling of ERK5 between the nucleus and cytoplasm.

Similar to ERK5-FL, ERK5-4xA_i_ and ERK5-T732A mutants preferentially localized in the cytoplasm (Figure 2A,B). In contrast, mimicking phosphorylation at the C-terminal tail caused a notable increased proportion of ERK5 in the nucleus (Figure 2A,B). Interestingly, we found no significant advantage of multiple phosphorylation at Ser^706^, Thr^732^, Ser^753^ and Ser^773^ versus single T732E substitution (Figure 2A,B; compare ERK5-4xE_i_ and ERK5-T732E). As expected, a small, but nonetheless significant, proportion of ERK5-FL moved in the nucleus of cells stimulated with EGF (Figure 2C,D). Likewise, we observed a slightly higher proportion of nuclear ERK5-T732E in EGF-treated compared to unstimulated cells (Figure 2C and D). On the contrary, Ala^732^ mutation blocked the nuclear translocation of ERK5 in response to EGF stimulation (Figure 2C,D). Together, these observations confirmed an important regulatory role of Thr^732^ phosphorylation in ERK5 nuclear shuttling.

### 2.3. Phosphorylation at Thr^732^ Enhances ERK5 Transcriptional Activity

Previous studies have found that mimicking phosphorylation at multiple sites in the C terminus was required for maximal ERK5 transcriptional activity [8,11,13]. To establish the specific requirement of Thr^732^ in ERK5-mediated transcription, we tested the ability of various ERK5 mutants to increase transcription using a MEF2-dependent luciferase reporter construct. We verified by immunoblot analysis that tetracycline induced expression of all mutants to a similar level for comparison (Figure 4). We found that induced expression of ERK5-FL or ERK5-ΔC (1-575) caused a small, nonetheless noticeable, increase in MEF2-luc activity (Figure 4A). We further analyzed the transcriptional activity of phosphodeficient forms of full-length ERK5, alongside two phosphomimetics in which Ser^706^, Thr^732^, Ser^753^ and Ser^773^ (ERK5-4xE_i_), or Thr^732^ alone (ERK5-T732E), were replaced by Glu residues. We observed that the phosphomimetics enhanced transcription by around 3-fold over the phosphodeficient mutants which displayed a similar activity as that of ERK5-FL or ERK5-ΔC (Figure 4A). In agreement with our previous observation (Figure 2A,B), we found no marked difference between the substitution of four Glu residues versus single Glu mutation at Thr^732^. The critical importance of phosphorylation at Thr^732^ was further demonstrated by evidence that enhanced MEF2-luciferase activity could not be produced by mimicking phosphorylation at three serine residues (Ser^706^, Ser^753^ and Ser^773^, or Ser^769^, Ser^773^ and Ser^775^) in the context of an unphosphorylatable Ala^732^ residue (Figure 4B; 3xE_i_-T732A and 3xE_ii_-T732A mutants).

Subsequently, we generated another set of T732A and T732E substitutions in a kinase-dead mutant form of ERK5 unable to bind ATP (D200A) [9], in order to dissociate the functional requirement of Thr^732^ phosphorylation from ERK5 catalytic activity (Figure 1A). We found that the loss of catalytic function blocked the two-fold increase in MEF2-luciferase activity caused by induced expression of ERK5-T732A (Figure 4C). In contrast, D200A mutation only partially reduced the transcriptional activity of ERK5-T732E, indicating that phosphorylation at Thr^732^ alone could induce transcription (Figure 4C). Remarkably, the C-terminal fragment of ERK5 (aa 411–816) was a more potent inducer of MEF2-luciferase activity than ERK5-FL or any of the nuclear ERK5-ΔC mutants comprising the kinase domain (Figure 4D). Collectively, these results indicated that the C-terminal tail of ERK5 functioned as a non-catalytic inducer of ERK5-mediated transcription. More specifically, we propose that phosphorylation at Thr^732^ enhances transcriptional activation, not only by allowing ERK5 to enter the nucleus, but through positive regulation of the TAD (aa 664-789). 

### 2.4. Phosphorylation at Thr^732^ Confers Binding Interaction Specificity 

To further understand the mechanism underpinning ERK5-mediated transcription, we utilized a mass spectrometry-based quantitative proteomics approach to globally investigate the impact of phosphorylation of the C-terminal half of ERK5 on molecular interactions and potentially identify binding partners implicated in genomic interactions. We compared M2-pull down purifications from ERK5-FL non-induced Flp-In HeLa cells (–Tetracycline) and from Flp-In HeLa cells induced (+Tetracycline) to express ERK5-FL or ERK5-T732E (Figure 5A).

We confirmed that the biological triplicates were similar with a Pearson correlation between 0.85 and 0.95 (Figure 5B). Moreover, although a similar number of proteins were detected in M2 immune complexes from induced and non-induced cells (Figure 5C), MEK5 and ERK5 were significantly enriched in the precipitates from induced conditions (Figure 5D). This initial analysis confirmed that we could utilize non-induced samples as negative controls to rule out non-specific binding interactions. Furthermore, consistent with the absence of stimulation, no phosphorylation was detected on the MAPK activation motif Thr^218^-Glu-Tyr^220^ in the N-terminal domain of ERK5. In contrast, we were able to identify phosphorylated ERK5-FL at Thr^732^, corroborating the biological significance of this phosphoacceptor site (Figure 5D). Interestingly, the phospho-Thr^732^ ERK5 peptide was also phosphorylated at Ser^730^. This site is the second most phosphorylated C-terminal site after Thr^732^ found in unbiased high through put phosphoproteomic screens (https://www.phosphosite.org/homeAction.action). Together, these observations confirmed with high confidence that our methodology was robust and the bioinformatics analysis of the data was highly reliable for discovering novel and relevant ERK5 binding partners.

Hierarchical clustering was subsequently performed on 2368 proteins associated with immune complexes (Figure 6A and Appendix A). Proteins interacting selectively with ERK5-FL were found in cluster 1. Cluster 2 included proteins exhibiting similar binding interaction with ERK-FL and ERK5-T732E and cluster 3 was comprised of proteins exhibiting enhanced affinity for ERK5-T732E. Overall, ERK5-FL had significantly more interacting partners (231) than ERK5-T732E (45). Interestingly, the most significantly enriched interactor identified in the ERK5-FL after MEK5 was the scaffold protein SHC1 which links cell surface growth factor receptors to signaling pathways. Moreover, several proteins involved in metabolism, e.g., UGDH, ATP5A1, NAMPT, ATP6V1H, PHGDH, ALDH, were reproducibly detected with higher signal intensities in ERK5-FL than in ERK5-T732 pull downs. In contrast, various histones and several small nuclear ribonucleoproteins implicated in pre-mRNA splicing were highly represented in the top most enriched proteins detected in the M2 immune complexes isolated from ERK5-T732, compared with ERK5-FL, expressing HeLa cells. Functional network analysis within each cluster was subsequently performed in STRING and visualized with Cytoscape to reveal more global differences between ERK5-FL and ERK5-T732 immunoprecipitates (Figure 6B–D). Together, these findings demonstrated for the first time that phosphorylation at Thr^732^ affected selective binding affinity of ERK5 with specific downstream partners.

## 3. Discussion

This study aimed at elucidating the functional relevance of post-translational modification of the C terminus of ERK5. We identified Thr^732^ as a gatekeeper residue involved in controlling C-terminal-mediated nuclear translocation and in enhancing transcription. Importantly, the detection of Thr^732^ phosphorylation by mass spectrometry indicated that this phosphoacceptor site was biologically relevant. We propose that single phosphorylation at Thr^732^ stabilizes ERK5 in an open conformation where the NLS is exposed to facilitate nuclear entry. This is functionally important given that T732A mutation blocked the nuclear import of ERK5 caused by activating phosphorylation of ERK5 by MEK5 in response to EGF stimulation. Previous studies had observed that mimicking C-terminal phosphorylation increased ERK5 transcriptional activity through AP-1 and Nur77 [8,11]. The finding that phosphorylation at Thr^732^ was a more potent inducer of MEF2 activity than any of the nuclear ERK5-ΔC mutants provided further compelling evidence supporting the functional requirement of the TAD. Accordingly, we previously found that ectopic expression of ERK5-ΔC (1−575) failed to enhance STAT3 activity to the same level as that observed with overexpressing ERK5-FL [19]. 

To further understand the mechanism underpinning the transcriptional function of the C-terminal tail, we searched potential ERK5 binding partners by utilizing a mass spectrometry-based quantitative proteomics approach. Interestingly, no transcription factors known to be phosphorylated by ERK5 were detected in our proteomic screen of ERK5 immunecomplexes. This may not be surprising given that enzyme-substrate complexes are known to be very short lived. Nonetheless, we discovered that Thr^732^ phosphorylation increased the affinity of ERK5 for histones. Interestingly, MAPKs have been shown to interact with chromatin and chromatin-associated substrates [20]. For example, ERK2 was found directly interacting with DNA [21], p38 associated with chromatin remodelling complexes (SWI/SNF) [22] and JNK in complex with the acetyltransferase complex (ATAC) [23]. Therefore, we postulated that Thr^732^ phosphorylation might enhance transcription by facilitating chromatin association of ERK5, thereby bringing the TAD in the vicinity of promoter regions.

Accordingly, we found a clear enrichment of chromatin-bound ERK5-T732E over ERK-FL and ERK5-ΔC (data not shown). However, an initial chromatin immunoprecipitation experiment followed by high throughput sequencing (ChIP-Seq) failed to convincingly demonstrate that ERK5 interacted with the genome. Interestingly, the yeast homolog of ERK5, Mpk1, allowed transcriptional elongation of stress-induced genes by preventing prematured termination through direct interactions with the RNA polymerase associated factor 1 (PAF1) complex at promoters and coding regions of target genes [24]. Likewise, ERK5 in a complex with cofilin, an actin-severing protein required for actin cytoskeleton reorganization, facilitated PAF1 recruitment at genomic loci to permit estrogen-induced expresion of genes associated with breast cancer cell proliferation [25]. Although, we did not detect any interaction of ERK5 with PAF1, ERK5-T732E exhibited increased affinity for a number of small nuclear ribonucleoproteins involved in pre-mRNA splicing and components of the RNA polymerase II holoenzyme. Together, these findings suggested that phosphorylation at Thr^732^ might enhance gene expression by enabling mRNA elongation at specific genomic loci. 

The demonstration that XMD8-92 treatment blocked the appearance of the slow migrating ERK5 band caused by EGF stimulation was consistent with the idea that C-terminal phosphorylation could occur as a consequence of auto-phosphorylation following ERK5 activation [8,17]. Nonetheless, we found that the Ser^730^-Glu-Thr^732^-Pro motif in the C-terminal region was phosphorylated in the absence of phosphorylation at Thr^218^-Glu-Tyr^220^ in the activation loop, thereby demonstrating that C-terminal phosphorylation could also occur independently of MEK5 [11,12,13,16]. Interestingly, Thr^732^ has recently been implicated in cross-talk signaling mechanisms between ERK1/2, CDK1 and ERK5 during mitosis and in response to oncogenic RAS and BRAF activation [11,12,13,16]. Collectively, these studies highlighted the potential clinical relevance of C-terminal phosphorylation in melanoma and HER2 over-expressing breast cancer growth. Therefore, future work should focus on understanding the interdependency of Ser^730^ and Thr^732^ phosphorylation and test the impact of dual phosphorylation at these residues on nuclear localization and transcriptional function, in the context of tumorigenesis and cancer resistance to therapy. These studies will require the generation of additional mutants in which Ser^730^ is replaced by Ala or Glu residues together with T732E and T732A mutations. Additionally, D200A kinase-dead mutation and the utilization of ERK1/2 or CDK1 inhibitors will aid establishing the requirement of ERK5 activation for autophosphorylating its tail. Overall, we anticipate that understanding the mechanism responsible for phosphorylation of the Ser^730^-Glu-Thr^732^-Pro motif might, not only advance our molecular understanding of ERK5-mediated signaling, but also reveal potentially important novel cancer targeted strategies based on blocking the oncogenic function of ERK5. 

## 4. Materials and Methods 

### 4.1. Generation of Isogenic Tet-Inducible ERK5 Expressing HeLa Cell Lines

N-terminal Flag epitope-tagged full length (human) ERK5 (ERK5-FL) cDNA [3], and C- and N-terminal truncated mutants were sub-cloned into a pCDNA5/FRT/TO vector (ThermoFisher Scientific, Waltham, MA, USA) modified to contain a 5′ Lamin UTR using Not1 and BamH1 digestion [26]. The various ERK5 phosphomutants were created by site-directed mutagenesis (QuikChange, Agilent technologies, Santa Clara, CA USA) and sub cloned into the pCDNA5/FRT/TO vector (Table 1). Primer sequences are indicated in (Table 2). All constructs were confirmed by sequencing. The pcDNA5/FRT/TO expression vectors containing ERK5-FL or ERK5 mutants were subsequently co-transfected with pOG44 into the Flp-In Hela host cell line using the jetPEI transfection reagent according to manufacturer’s instructions (Polyplus, Illkirch, France). The Flp recombinase expressed from pOG44 catalyzed homologous recombination between the FRT sites in the host cells and the pcDNA5/FRT/TO expression vectors. Stable recombinant Flp-In Hela cells were selected in hygromycin (2 mg/mL, ThermoFisher Scientific) for 2 weeks. Resistant colonies were pooled and expanded in DMEM supplemented with 10% fetal bovine serum (FBS, Sigma-Aldrich, Dorset, UK), 1% penicillin/streptomycin (Sigma-Aldrich). Expression of the constructs was induced by incubating the cells with 1 ug/mL tetracycline (Sigma-Aldrich) for 24 h prior to being harvested. 

### 4.2. Immunoblot Analysis

Proteins were extracted from cells in radioimmunoprecipitation assay (RIPA) buffer (Sigma) containing EDTA-free protease inhibitor cocktail (Roche, Basel, Switzerland). Extracts (50 μg) were resolved by SDS-PAGE and electrophoretically transferred to an Immobilon-P membrane (Merck Millipore, West Lothian, UK). The membranes were saturated in 3% nonfat dry milk and probed overnight at 4 °C with a primary antibody (1:2000) to the Flag-epitope (M2, Sigma-Aldrich). Anti-β actin (1:2000, Sigma-Aldrich) and anti-vinculin (1:2000, Abcam, Cambridge, UK) antibodies were used to monitor protein loading. Immunocomplexes were detected by enhanced chemiluminescence with IgG coupled to horseradish peroxidase as the secondary antibody (GE Healthcare, Little Chalfont, UK). 

### 4.3. Luciferase Reporter Assays

The MEF2 reporter luciferase plasmid [28] was transiently transfected into Flp-In HeLa cell lines. A pRL-Tk plasmid encoding Renilla luciferase was co-transfected to monitor transfection efficiency. Aliquots of cell lysates were assayed for firefly and *Renilla* luciferase activities using the Dual-luciferase reporter assay kit (Promega, Wisconsin, UK) on an Orion microplate luminometer. 

### 4.4. Immunofluorescence

Flp-In HeLa cell lines cultured on glass coverslips for 72 h in presence of tetracycline were fixed in 4% paraformaldehyde for 20 min and permeabilized for 25 min with 0.1% Triton X-100 in PBS. After saturation with 1% BSA for 1 h, cells were incubated with the M2 antibody (1:200) for 1 h at room temperature. After 3 washes with PBS, the cells were exposed to a secondary antibody (1:1000) conjugated to Alexa-488 (ThermoFisher Scientific) in addition to Phalloidin (Sigma-Aldrich) for 1 h. Nuclei were stained with DAPI (ThermoFisher Scientific). Fluorescence images were viewed with a Z1 inverted Axio Observer microscope from ZEISS. Fluorescent intensity of nuclear and cytoplasmic signals was quantified using ImageJ software. 

### 4.5. Mass Spectrometry and Raw Data Analysis 

Proteins extracted in RIPA (10 mg) were precleared with an anti-mouse IgG (10 μg/mg; Sigma-Aldrich) supplemented with Protein G-Sepharose beads (ThermoFisher Scientific) and inversion rotated for 5 h at 4 °C. After centrifugation at 1000× *g* for 5 min, cleared lysates were incubated with an anti-Flag M2 affinity gel (Sigma-Aldrich) (10 μg/mL) for 90 min at 4 °C. Immune complexes were washed three times with ice-cold lysis buffer supplemented with 500 nM NaCl, followed by two washes with 150 nM NaCl and analyzed by SDS-PAGE. The gel was then digested with trypsin, as previously described [29]. Digested samples were analyzed by LC-MS/MS using an UltiMate^®^ 3000 Rapid Separation LC (RSLC, Dionex Corporation, Sunnyvale, CA, USA) coupled to a QE HF (ThermoFisher Scientific) mass spectrometer. Mobile phase A was 0.1% formic acid in water and mobile phase B was 0.1% formic acid in acetonitrile and the analytical column utilized was a 75 mm × 250 μm inner diameter, 1.7 μm CSH C18 (Waters). Specifically, purified samples were concentrated and desalted on C18 StagTips, as previously described [28] and transferred to a 5 μL loop before loading on to the column at a flow of 300 nL/min for 5 min at 5% B. The loop was subsequently taken out of line and the peptides separated using a gradient that went from 5% to 7% B and from 300 nL/min to 200 nL/min in 1 min followed by a shallow gradient from 7% to 18% B in 64 min, then from 18% to 27% B in 8 min and finally from 27% B to 60% B in 1 min. The column was washed at 60% B for 3 min before re-equilibration to 5% B in 1 min. At 85 min the flow was increased to 300 nL/min until the end of the run at 90 min. Mass spectrometry data was acquired in a data dependent manner for 90 min in positive mode. Peptides were selected for fragmentation automatically by data dependent analysis on a basis of the top 12 peptides with m/z between 300 to 1750 Th and a charge state of 2, 3 or 4 with a dynamic exclusion set at 15 s. The MS Resolution was set at 120,000 with an AGC target of 3e6 and a maximum fill time set at 20 ms. The MS2 Resolution was set to 30,000, with an AGC target of 2e5, a maximum fill time of 45 ms, isolation window of 1.3Th and a collision energy of 28.

Raw data were analyzed with the MaxQuant software suite, version 1.5.6.5, with the integrated Andromeda search engine [30]. Proteins were identified by searching the HCD-MS/MS peak lists against a target/decoy version of the human Uniprot database, which consisted of the complete proteome sets and isoforms (2016 release) supplemented with commonly observed contaminants such as porcine trypsin and bovine serum proteins. Tandem mass spectra were initially matched with a mass tolerance of 7 ppm on precursor masses and 0.02 Da or 20 ppm for fragment ions. Cysteine carbamidomethylation was searched as a fixed modification. Protein N-acetylation, oxidized methionine and deamidation of asparagine and glutamine were searched as variable modifications. Label-free parameters were used as described [31]. False discovery rate was set to 0.01 for peptides, proteins and modification sites. Minimal peptide length was six amino acids. The dataset were filtered by posterior error probability to achieve a false discovery rate below 1% for peptides and proteins. Only peptides with Andromeda score >40 were included. 

### 4.6. Data Analysis 

At least two unique peptides and a protein-sequence coverage of 5% were required. Intensity values were log2 transformed and normalized using the function “normalizeQuantiles” included in the LIMMA package of Bioconductor in R ([32] and http://www.ncbi.nlm.nih.gov/pubmed/16646809). All values were centered to the mean. The software Perseus was used for integrating missing values with normal distribution and for hierarchical clustering using default options [33]. The overview of all the proteins in the three clusters was based on STRING database [34] and visualized in Cytoscape depending on the MCODE plug-in ((https://cytoscape.org/).

### 4.7. Statistical Analysis

To compare values between multiple test groups we performed a one-way ANOVA followed by Tukey’s test. Data were analyzed using Prism software (GraphPad, San Diego, USA). The quantitative interactomics experiment was performed in triplicates.

### 4.8. Data Availability

The authors declare that all data supporting the findings of this study are available within the article and its supplementary information files or from the corresponding authors on reasonable request. The mass spectrometry proteomics data have been deposited to the ProteomeXchange Consortium via the PRIDE partner repository with the dataset identifier PXD014028. Reviewer account details: *Project Name:* A gatekeeper residue controls transcriptional activity of the extracellular signal-regulated protein kinase 5 (ERK5) through molecular interactions with the C-terminal tail. *Username:* reviewer86943@ebi.ac.uk, *Password:* Oouljqxr.

## Figures and Tables

**Figure 1 ijms-21-00929-f001:**
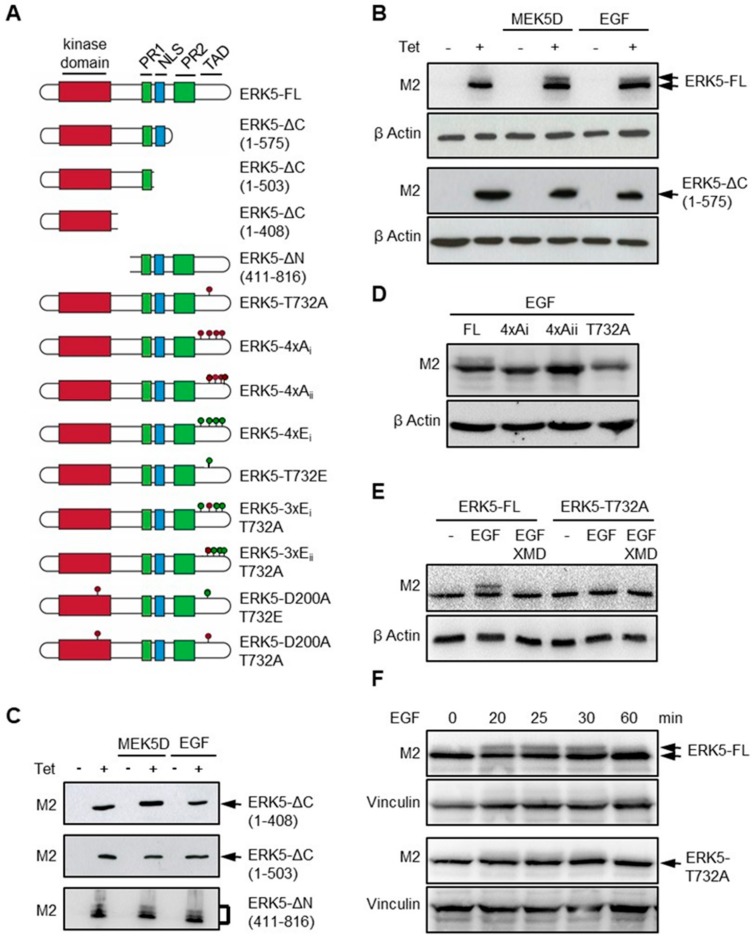
Extracellular signal-regulated protein kinase 5 (ERK5) phosphorylation at Thr^732^ is a pre-requisite for C-terminal phosphorylation. (**A**) A schematic illustration and nomenclature of ERK5-FL, truncated N-terminal (ERK5-ΔN) and C-terminal (ERK5-ΔC) mutants, and various mutants in which specific serine and/or threonine residues were replaced by alanines or glutamic acids. (**B****–F**) Recombinant Flp-In HeLa cells were incubated with Tet (+) for 24 h. Mock treated cells with DMSO (−) were used as controls. Where indicated, the cells were transfected with a construct encoding HA-tagged MEK5D 24 h before Tet induction, or stimulated for 25 min, unless indicated otherwise (panel F), with EGF (20 ng/mL) before harvesting. In panel E, cells were pre-incubated with XMD8-92 (5 μM) for 1 h prior to epidermal growth factor (EGF) stimulation. Cell lysates were analyzed by immunoblot using a specific antibody to the Flag-tagged epitope (M2). Antibodies to β actin or vinculin were used to monitor protein loading.

**Figure 2 ijms-21-00929-f002:**
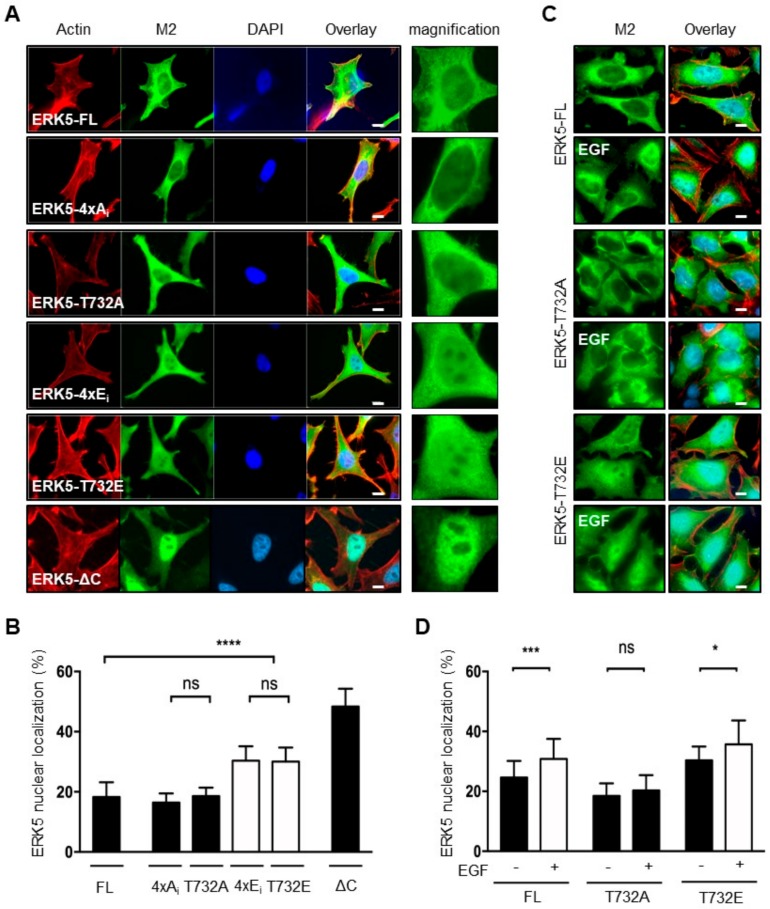
Phosphorylation at Thr^732^ promotes the nuclear translocation of ERK5. Recombinant Flp-In HeLa cell lines were grown on glass coverslips and incubated with tetracycline for 24 h before fixation. Where indicated, the cells were stimulated with EGF (20 ng/mL) for 25 min. (**A**,**C**) Subcellular localization of ectopically expressed ERK5-FL, ERK5-ΔC and specific phospho-deficient or phosphomimetics mutants were visualized using an antibody to the Flag-tagged epitope (M2, green). Phallodin staining (red) was used to detect actin. Nuclei were detected with DAPI (blue). Scale bars: 10 μM. (**B**,**D**) The immunofluorescent signal of the Flag epitope was quantitated with ImageJ. The data expressed as percent of signal in the nucleus correspond to the mean ± SD (*n* = 30 cells). * *p <* 0.05; *** *p < 0.001* and **** *p* < 0.0001 indicate significant differences. ns indicates no statistical difference.

**Figure 3 ijms-21-00929-f003:**
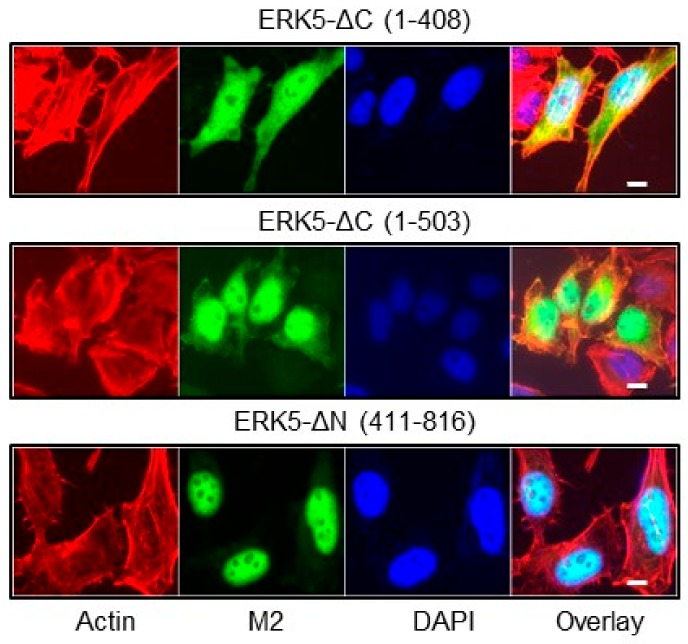
Subcellular localization of C- and N-terminal truncated ERK5 mutants. Recombinant Flp-In HeLa cell lines were grown on glass coverslips and incubated with tetracycline for 24 h before fixation. Subcellular localization of ectopically expressed ERK5-ΔC and ERK5-ΔN truncated mutants was visualized using an antibody to the Flag-tagged epitope (M2, green). Phallodin staining (red) was used to detect actin. Nuclei were detected with DAPI (blue). Scale bars: 10 μM.

**Figure 4 ijms-21-00929-f004:**
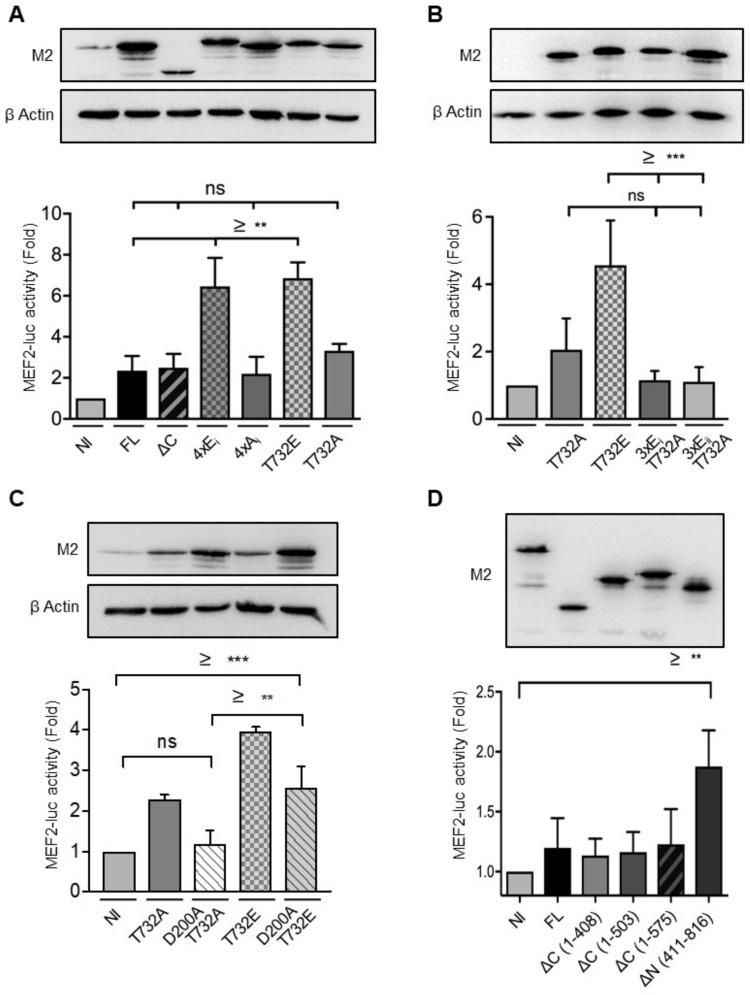
Phosphorylation at Thr^732^ enhances ERK5-mediated transcription. Recombinant Flp-In HeLa cell lines were transfected with a construct encoding a MEF2 luciferase reporter. (**A–D**) 24 h later, the cells were incubated with tetracycline for 48 hours to induce expression of ERKFL, ERK5-ΔC and ERK5-ΔΝ fragments, or specific phospho-deficient or phosphomimetics mutants, as indicated. Non-induced (NI) cells were used as controls. Efficiency of transfection was controlled by co-transfecting a *Renilla* firefly encoding construct. Immunoblot analyses of the cell lysates demonstrate similar level of expression of ERK5-FL and the various mutants. The MEF2 luciferase activity normalized to that of *Renilla* luciferase is expressed as fold to compare relative transcriptional activity under basal condition. The data represent the mean ± SD of three independent experiments performed in duplicate. *** p* < 0.01 and **** p* < 0.001 indicate significant differences. ns indicates no statistical difference.

**Figure 5 ijms-21-00929-f005:**
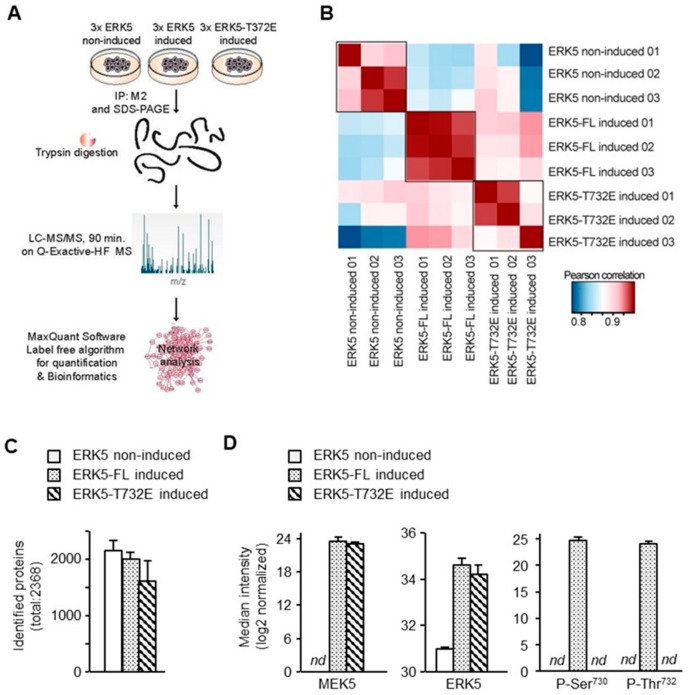
Quantitative mass spectrometry analysis of M2 immunecomplexes. (**A**) Illustration of the workflow to analyze the interactomes of non-induced, ERK5-FL and ERK5-T732E pull downs. Recombinant Flp-In HeLa cell lines were lysed in RIPA buffer 24 h after tetracycline induction. Mock treated cells with DMSO were utilized as non-induced controls. (**B**) Pearson correlation coefficients demonstrating the reproducibility of pull downs analyzed by mass spectrometry in the different conditions. (**C**) Quantification of mean total number of proteins in each pull down. (**D**) Signal intensities of MEK5 and ERK5 proteins, and phosphorylated Ser^730^ and Thr^732^ residues, detected by mass spectrometry in the non-induced and induced ERK5-FL, or ERK5-T732E precipitates. The data represent the median normalized intensity values ± SD of three independent experiments. *nd* indicates that the protein or the phosphorylated residue was not-detectable in the corresponding immunoprecipitate.

**Figure 6 ijms-21-00929-f006:**
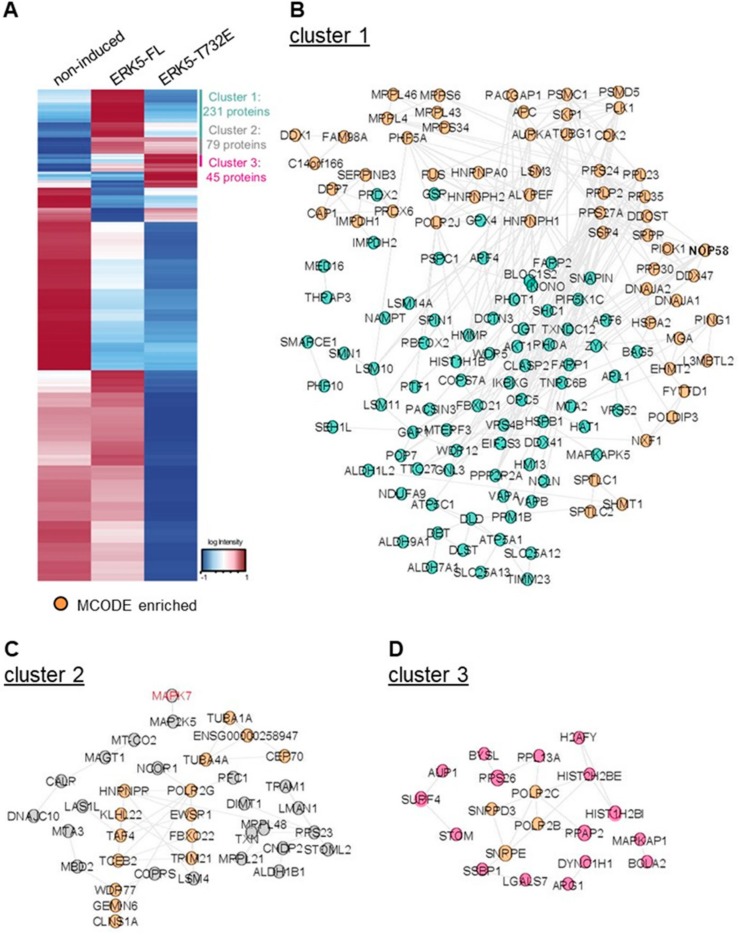
Hierarchical clustering demonstrates distinct interactomes for ERK5-FL and ERK5-T732. Hierarchical clustering was performed on 2368 proteins detected in immune complexes using the software Perseus. (**A**) The dendrogram shows clustering of proteins mostly enriched in ERK5-FL (cluster 1), in ERK5-T732E (cluster 3), or in both (cluster 2) immune complexes. (**B**–**D**) The proteins in each cluster were visualized in Cytoscape after analysis in the STRING software to provide known interactions among proteins. MCODE enrichment was utilized to identify more interconnected nodes, i.e., groups of proteins exhibiting higher probability to interact with each other.

**Table 1 ijms-21-00929-t001:** ERK5 and MEK5 constructs utilized in this study.

Mutant	Details	Reference
ERK5-FL		[3]
ERK5-ΔC	Deletion of amino acids 576-816 (-TAD)	[9,10]
ERK5-ΔC (1-503)	Deletion of amino acids 504-816 (-PR2)	[4]
ERK5-ΔC (1-408)	Deletion of amino acids 409-816 (-NLS)	[18]
ERK5-ΔN (411-816)	Deletion of amino acids 1-410	
ERK5-T732A	Substitution of Thr^732^ to Ala	[8,11,12,13,17]
ERK5-T732E	Substitution of Thr^732^ to Glu	[8,11,12,13,17]
ERK5-4xA_i_	Substitution of Ser^706^, Thr^732^, Ser^753^, Ser^773^ to Ala	[11]
ERK5-4xA_ii_	Substitution of Thr^732^, Ser^769^, Ser^773^, Ser^775^ to Ala	[8]
ERK5-4xE	Substitution of Ser^706^, Thr^732^, Ser^753^, Ser^773^ to Glu	[11]
ERK5-4xE_ii_	Substitution of Thr^732^, Ser^769^, Ser^773^, Ser^775^ to Glu	[8]
ERK5-3xE_i_-T732A	Substitution of Ser^706^, Ser^753^, Ser^774^ to Glu and Thr^732^ to Ala	[11]
ERK5-3xE_ii_-T732A	Substitution of Ser^769^, Ser^773^, Ser^775^ to Glu and Thr^732^ to Ala	[8]
ERK5-D200A-T732E	Substitution of Asp^200^ to Ala and Thr^732^ to Glu	[9]
ERK5-D200A-T732A	Substitution of Asp^200^ to Ala and Thr^732^ to Ala	[9]
MEK5DA	Substitution of Ser^311^ and Thr^315^ to Asp	[27]

**Table 2 ijms-21-00929-t002:** Primer sequences used for mutagenesis and subcloning.

ERK5 mutant	Sequence
ERK5-FL	GAGAGGATCCATGGCCGAGCCTCTG GAGAGCGGCCGCTCAGGGGTCCTGGAG
ERK5- ΔC	GAGAGGATCCATGGCCGAGCCTCTG GAGAGCGGCCGCGGCCATTCGAGTCCA
ERK5-ΔC (1-503)	GAGAGGATCCATGGCCGAGCCTCTG GAGAGCGGCCGCAGCCACAGGCTG
ERK5-ΔC (1-408)	GAGAGGATCCATGGCCGAGCCTCTG GAGAGCGGCCGCAGGCTCAGGAGC
ERK5-ΔN (411-816)	GAGAGGATCCGGCTGTCCAGAT GAGAGCGGCCGCTCAGGGGTCCTGGAG
ERK5-T732A	CACTGCCCTTTGGTGCGCCTGAGAACACAGG
ERK5-T732E	CCCCACTGCCCTTTGGCTCGCCTGAGAACACAGGG
ERK5-4xA_i_	CACTGCCCTTTGGTGCGCCTGAGAACACAGG CAAGCAGGGAGGCTGCGAGAGAGGCTGAATC CCTGTGTTCTCAGGCGCACCAAAGGGCAGTG GATTCAGCCTCTCTCGCAGCCTCCCTGCTTG
ERK5-4xA_ii_	CAGCAAGCAGGGCGGCTGCGAGAGAGGCTGA CAGGATGGCCAGGCAGATGCAGCCTCTCT CACTGCCCTTTGGTGCGCCTGAGAACACAGG
ERK5-4xE	GTCAGCAAGCAGGGAGGCCTCGAGAGAGGCTGAATCTGC ATCAGCCACGCCCATGTCGAACTCCTGGTTTAAGAATTCCTCCAG CCCTGTGTTCTCAGGCGAGCCAAAGGGCAGTGGGG CCCCACTGCCCTTTGGCTCGCCTGAGAACACAGGG
ERK5-4xE_ii_	CCCCACTGCCCTTTGGCTCGCCTGAGAACACAGGG CCAGTCAGCAAGCAGCTCGGCCTCGAGAGAGGCC TCATCTGCCTGGCCATCCTGTGGCCC
ERK5-3xE_i_-T732A	CACTGCCCTTTGGTGCGCCTGAGAACACAGG GTCAGCAAGCAGGGAGGCCTCGAGAGAGGCTGAATCTGC ATCAGCCACGCCCATGTCGAACTCCTGGTTTAAGAATTCCTCCAG CCCTGTGTTCTCAGGCGAGCCAAAGGGCAGTGGGG
ERK5-3xE_ii_-T732A	CACTGCCCTTTGGTGCGCCTGAGAACACAGG CCAGTCAGCAAGCAGCTCGGCCTCGAGAGAGGCC TCATCTGCCTGGCCATCCTGTGGCCC
ERK5-D200A-T732E	CACGAGCCATACCAAAGGCACCAATCTTGAGCTCA CCCCACTGCCCTTTGGCTCGCCTGAGAACACAGGG
ERK5-D200A-T732A	CACGAGCCATACCAAAGGCACCAATCTTGAGCTCA CACTGCCCTTTGGTGCGCCTGAGAACACAGG

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
