# Peer review of "Discovery of a Gatekeeper Residue in the C-Terminal Tail of the Extracellular Signal-Regulated Protein Kinase 5 (ERK5)"

_ijms, 2020, doi:10.3390/ijms21030929_

Round 1

Reviewer 1 Report

This an excellent manuscript detailing the gatekeeper residue and the importance of phosphorylation of Thr732 in promoting binding interactions of the protein and its Nuclear localization. This adds to the work done by Honda etal (2015 PLoS One) in studying the phosphorylation of the same amino acid.

Adam Pearson et al have done a commendable job in working thoroughly on the importance of phosphorylation of Thr 732 in activating ERK5. A large number of truncated and modified mutant studies have been performed. The work is very thorough. However one mmore experiment could further tighten the data proving the importance of Thr732 phosphorylation.

Major

The authors have thoroughly investigated replacing the Thr 732 with other amino acids and have proven that Thr 732 phosphorylation is important for activation of ERK5 as well as its nuclear localization. If Thr is replaced by serine will ERK5 still get activated? Thr has an extra methyl group compared to Ser. In that case the specificity of Thr phosphorylation will get stressed further. It could be proven that it is not the site alone that is important, it is both the site and the amino acid.

I would suggest the authors perform a set of experiments replacing Thr 732 with Ser.

Minor comment

Line 180 which lack NLS remain nuclear. NLS if absent should be cytoplasmic and not nuclear. Please clarify this.

Author Response

Major comment

The reviewer agree that we have provided strong evidence that phosphorylation at Thr732 is critical for controlling the nuclear translocation of ERK5 and ERK5-mediated transcription. They suggest creating an additional mutant in which Thr732 is replaced by a serine residue to address the requirement of residue specificity at this position. This is an interesting question. However, given that the main objective of this study was to identify the functional consequence of post-translational modification of the C-terminal tail of ERK5, we argue that this issue should be addressed in a follow up study with the generation of a novel isogenic Tet-inducible ERK5 expressing HeLa cell line.

Minor comment

“Line 180: which lack NLS remain nuclear. NLS if absent should be cytoplasmic and not nuclear. Please clarify this.”

We agree that the nuclear localization of ERK5-DC (1-408) and ERK5-DC (1-503) (Figure 3) is puzzling given that these mutants lack the NLS. However, they also lack the nuclear export-dependent activity previously defined by an intermolecular interaction between the N- and C-terminal half of the protein. Based on this previous knowledge, we propose that the C-terminal tail of ERK5 might comprise a cytoplasmic retention function, possibly through protein association, that contributes to controlling the dynamic shuttling of ERK5 between the nucleus and cytoplasm. The text in the manuscript (P4L181) was modified accordingly.

.

Reviewer 2 Report

In this study, the authors found that ERK5 phosphorylation at Thr732 is responsible for the nuclear translocation and transcriptional enhancement. Furthermore, they showed by the proteomic analysis that phosphorylation of Ser730-Glu-Thr732-Pro motif was occurred independently of dual phosphorylation at TEY activation motif.

Overall, the experiments were performed properly and their findings are reliable. However, the most part of the manuscript (Figs.1-4) has already been demonstrated by other researchers previously (for example, references 8, 11 and 13). Their proteomic analysis is novel and interesting, but this result does not occupy the main part of this manuscript.

Major point:
As mentioned above
, I feel that the novelty of this manuscript is limited.

Minor points:
In Fig. 1E
, EGF caused the ERK5 band-shift, which was completely blocked by XMD8-92 and T732A mutant. This means that ERK5 activated by EGF (probably through MEK5) auto-phosphorylates the Thr732 residue which caused the band-shift. In contrast, other kinases such as ERK1/2 or CDK1 can phosphorylates ERK5 at Thr732 (references 11 and 13) . The authors says that no phosphorylation was detected on the ERK5 activation motif (TEY) whereas Thr732 phosphorylation was identified by the mass spectrometry.

So, please explain whether Thr732 is phosphorylated by auto-phosphorylation or other signaling pathway? It may not be essential, but experiments using ERK5 kinase-dead mutant (D200A), pERK5 (TEY) antibody, and ERK1/2 or CDK1 inhibitors may be useful to address the issue.

Author Response

Major comment

The reviewer “feel that the novelty of this manuscript is limited”. We disagree given that this study is the first to reveal convincing binding partners of ERK5 via quantitative mass spectrometry analysis. Moreover, our approach has unambiguously demonstrated that dual phosphorylation of the Ser730-Glu-Thr732-Pro motif is a biologically relevant phenomenon.

Minor point

The reviewer asked to clarify whether Thr732 is phosphorylated by auto-phosphorylation following ERK5 activation by MEK5 or via other signaling pathways. This issue was addressed accordingly by modifying the last paragraph of the discussion (p11L549).

Reviewer 3 Report

The authors investigated the significance of C terminal phosphorylation in the regulation of ERK5 function. Worth of note, the authors performed different types of experiments and provided number of data that are well linked in the discussion. The work is well-written and easy to follow in each step. 

Given the high scientific value of the manuscript, I suggest the publication of this manuscript in your journal.

Author Response

We thank the reviewer for their kind comment that “Given the high scientific value of the manuscript, I suggest the publication of this manuscript in your journal”.

Round 2

Reviewer 1 Report

I am satisfied with the explanation that the suggested experiment can be done as a follow up study